# An Unprecedented Tridentate-Bridging Coordination Mode of Permanganate Ions: The Synthesis of an Anionic Coordination Polymer—[Co^III^(NH_3_)_6_]_n_[(K(κ^1^-Cl)_2_(μ^2,2′,2″^-(κ^3^-O,O′,O″-MnO_4_)_2_)_n_^∞^]—Containing Potassium Central Ion and Chlorido and Permanganato Ligands

**DOI:** 10.3390/molecules29184443

**Published:** 2024-09-19

**Authors:** László Kótai, Kende Attila Béres, Attila Farkas, Berta Barta Holló, Vladimir M. Petruševski, Zoltán Homonnay, László Trif, Fernanda Paiva Franguelli, Laura Bereczki

**Affiliations:** 1Institute of Materials and Environmental Chemistry, HUN-REN Research Centre for Natural Sciences, H-1117 Budapest, Hungary; beres.kende.attila@ttk.hu (K.A.B.); homonnay.zoltan@ttk.elte.hu (Z.H.); trif.laszlo@ttk.hu (L.T.); ferpaivafran@gmail.com (F.P.F.); nagyne.bereczki.laura@ttk.hu (L.B.); 2György Hevesy PhD School of Chemistry, ELTE Eötvös Loránd University, H-1117 Budapest, Hungary; 3Department of Organic Chemistry and Technology, Faculty of Chemical Technology and Biotechnology, Budapest University of Technology and Economics, H-1111 Budapest, Hungary; farkas.attila@vbk.bme.hu; 4Faculty of Sciences, University of Novi Sad, 21000 Novi Sad, Serbia; hberta@uns.ac.rs; 5Faculty of Natural Sciences and Mathematics, Ss. Cyril and Methodius University, MK-1000 Skopje, North Macedonia; vladimirpetrusevski@yahoo.com; 6Institute of Chemistry, ELTE Eötvös Loránd University, H-1117 Budapest, Hungary; 7Chemical Crystallography Research Laboratory, Centre for Structural Science, HUN-REN Research Centre for Natural Sciences, H-1117 Budapest, Hungary

**Keywords:** chlorido and permanganato ligands, hexaamminecobalt(III), coordination polymer, tridentate permanganate, crystal structure, hydrogen bond, resonance Raman effect, cobalt manganite spinel, thermal decomposition, nanocrystallites

## Abstract

A unique compound (compound **1**) with structural features including an unprecedented tridentate-bridging coordination mode of permanganate ions and an eight-coordinated (rhombohedral) κ^1^-chlorido and tridentate permanganato ligand in a potassium complex containing coordination polymer (Co^III^(NH_3_)_6_]_n_[(K(κ^1^-Cl)_2_(μ^2,2′,2″^-(κ^3^-O,O′,O″-MnO_4_)_2_)_n_^∞^) with isolated regular octahedral hexamminecobalt(III) cation was synthesized with a yield of >90%. The structure was found to be stabilized by mono and bifurcated N-H∙∙∙Cl and N-H∙∙∙O (bridging and non-bridging) hydrogen bonds. Detailed spectroscopic (IR, far-IR, and Raman) studies and correlation analysis were performed to assign all vibrational modes. The existence of a resonance Raman effect of compound **1** was also observed. The thermal decomposition products at 500 °C were found to be tetragonal nano-CoMn_2_O_4_ spinel with 19–25 nm crystallite size and KCl. The decomposition intermediates formed in toluene at 110 °C showed the presence of a potassium- and chloride-containing intermediates combined into KCl during aqueous leaching, together with the formation of cobalt(II) nitrate hexahydrate. This means that the Co^III^–Co^II^ redox reaction and the complete decomposition of the permanganate ions occurred in the first decomposition step, with a partial oxidation of ammonia into nitrate ions.

## 1. Introduction

Reactive transition metal complexes containing reducing ligands such as ammonia, pyridine, or urea and oxoanions of metals can easily be transformed into amorphous oxides in solid-state quasi-intramolecular redox reactions under controlled heating, leading to the synthesis of various catalysts with special structures, surfaces, and catalytic properties [1,2,3,4,5,6,7,8,9,10,11,12,13,14,15,16,17,18,19,20,21,22]. Among these complexes, the permanganate salts of complex cations containing iron or cobalt have enormous importance in green fuel production technologies due to the activity of the prepared mixed oxides, for example in CO_2_ transformation reactions [23,24,25,26,27,28,29,30,31,32,33,34] and Fischer-Tropsch synthesis [21,35,36,37,38,39,40,41,42,43]. Not just reduction reactions but also oxidation reactions can be performed with a cobalt-manganese oxide catalyst in different procedures [44,45], like the oxidation of CO [46,47,48,49,50], volatile organic compounds (VOCs) [51,52,53] like propane [54,55,56], benzene [57], toluene [58,59,60], or other open chain [61,62,63] or cyclic [64,65] organic compounds, and special compounds like 5-hydroxymethylfurfural [66,67]. Furthermore, Co-Mn mixed oxides can be also applied in electrochemical [68,69,70,71,72,73,74] or inorganic material transformation [75,76,77,78] processes.

Cobalt complexes with ammonia ligands ensure high variability of the Co to Mn ratio in the prepared mixed oxides via the preparation of precursors with various Co to Mn ratios (1:1 for [Co(NH_3_)_4_CO_3_](MnO_4_) [3], [Co(NH_3_)_6_)]Cl_2_(MnO_4_) [42], [Co(NH_3_)_6_]Br_2_(MnO_4_) [38], 1:2 for [Co(NH_3_)_5_Cl](MnO_4_)_2_ [41] and 1:3 for [Co(NH_3_)_6_](MnO_4_)_3_ [79]). Furthermore, the final Co to Mn ratio also depends on the leaching conditions of the primary decomposition products before completing the formation of a spinel-type mixed oxide by post-heat treatment [41,42]. The precursors, thermal treatments, and processing of the primarily found amorphous oxides revealed various (Co,Mn)^T−4^(Co,Mn)_2_^OC−6^O_4+δ_ spinels having variable Co to Mn overall ratios with various cation (Co^II,III^ and Mn^II,III^) distributions between the T−4 and OC−6 spinel sites and with a variable number of defects. These conditions led to new catalysts with defect structures and different valence distributions of each metal at the T−4 and OC−6 locations [41,42]; this feature has a primary influence on their catalytic activity in various chemical processes [18,19,20]. The role of alkali promoters in the Fischer-Tropsch synthesis and other similar processes has been well-known for a long time [80,81,82]. Therefore, we prepared a new precursor, which is a triple salt containing potassium, cobalt, and permanganate ions.

We selected a poorly characterized potassium-containing triple salt of hexaamminecobalt(III) cations, chloride anions, and permanganate anions (K[Co(NH_3_)_6_]Cl_2_(MnO_4_)_2_, compound **1**), synthesized first by Klobb in 1887 [83] in the reaction of hexaamminecobalt(III) chloride (luteocobaltic chloride) and potassium permanganate as a by-product during the preparation of [Co(NH_3_)_6_]Cl_2_MnO_4_ (compound **2**) and [Co(NH_3_)_6_](MnO_4_)_3_ (compound **3**). Klobb recognized that during his analysis, ca. 20–30% of the ammonia was oxidized [84]. To study the structure and properties of compound **1**, including its suitability to prepare K-doped cobalt manganese oxides in solid phase low-temperature (<200 °C) quasi-intramolecular redox reactions, we optimized the preparation conditions, determined its structure, and studied its spectroscopic and thermal properties.

## 2. Results and Discussion

The purple beautiful hexagonal plates synthesized first by Klobb in 1887 in the reaction of hexaamminecobalt(III) chloride and potassium permanganate during the preparation of [Co(NH_3_)_6_](MnO_4_)_3_ and [Co(NH_3_)_6_]Cl_2_(MnO_4_) [84] were found to be the same compound—[Co(NH_3_)_6_][K(κ^1^-Cl)_2_(μ^2,2′,2″^-(κ^3^-O,O′,O″-MnO_4_)_2_)] (compound **1**)—as that prepared by us now (ESI Appendix A). Klobb found that this compound formed only if potassium permanganate was not used in excess to isolate [Co(NH_3_)_6_](MnO_4_)_3_. When Klobb mixed a cold concentrated solution of 2 moles of luteocobaltic chloride ([Co(NH_3_)_6_]Cl_3_) and 3 moles of KMnO_4_, a violet salt was gradually crystallized which could be dried at 50 °C. The same compound was formed when a concentrated solution of KCl was mixed with luteocobaltic chloride permanganate ([Co(NH_3_)_6_]Cl_2_(MnO_4_), compound **2**) or luteocobaltic permanganate ([Co(NH_3_)_6_](MnO_4_)_3_, compound **3**) with the formation of luteocobaltic chloride (Equation (1)) or KMnO_4_ (Equation (2)), respectively.
2[Co(NH_3_)_6_](MnO_4_)Cl_2_ + KCl → [Co(NH_3_)_6_]Cl_3_ + [Co(NH_3_)_6_][K(MnO_4_)_2_Cl_2_](1)
[Co(NH_3_)_6_](MnO_4_)_3_ + 2KCl → [Co(NH_3_)_6_][K(MnO_4_)_2_Cl_2_] + KMnO_4_(2)

It is obvious that compound **1** forms in a crystallization segment of the K,Co(NH_3_)_6_/MnO_4_,Cl—H_2_O system when the conditions are unfavorable for the crystallization of compounds **2** and **3**, or [Co(NH_3_)_6_]Cl_3_ and KMnO_4_. We optimized the preparation conditions with an increase of the reaction temperature to 60 °C (above this temperature the ammonia ligands of hexaamminecobalt(III) chloride reacted with potassium permanganate), keeping the molar ratio of [Co(NH_3_)_6_]Cl_3_ to KMnO_4_ at 1.5. With crystallization of the mother liquor overnight at room temperature, the yield increased to 91.6% with the formation of phase-pure compound **1**. The same anhydrous compound formed when the crystallization temperature of the mother liquor was +5 °C (ESI Appendix A).

The triple salt appears as small platelets, which are purple or black depending on the thickness, often with a greasy luster. Compound **1** is soluble in water (4.92 g/L) but with partial decomposition. The pH of the saturated solution is 5.9, which unambiguously shows that the complex cation minimally dissociates, with the liberation of ammonia ligands, due to the lack of ammonia protonation. The high electrolytic conductivity (4.45 mS), taking into account the lack of highly mobile hydroxide ions, may be attributed to the incongruent character of the dissolution and dissociation of the polymer anions with the formation of potassium, chloride, and permanganate anions. The electrolytic conductivities of hexaamminecobalt(III) chloride (μ = 3.41 mS, pH = 4.43), potassium permanganate (μ = 1.341 mS), and potassium chloride (μ = 2.08 mS) at equivalent concentrations of [Co(NH_3_)_6_]^3+^, chloride, potassium, and permanganate ions in the saturated aq. solution of compound **1** confirmed this assumption. Klobb found that the evaporation of the aqueous solution of compound **1** to dryness resulted in the formation of [Co(NH_3_)_6_]Cl_3_, [Co(NH_3_)_6_](MnO_4_)_3_, and KCl [84]. Our studies showed that pure compound **1** can be recrystallized from water without decomposition. Dry compound **1** is stable as a solid in the dark for more than 2 weeks at room temperature but explodes on fast heating or if smashed by a hammer.

### 2.1. Single Crystal X-ray Studies of Compound ***1***

Violet hexagonal prism-like single crystals of compound **1** were grown from water by slow partial evaporation at room temperature. Single crystal X-ray measurements were performed at 100 K and 273 K; the main crystal data and details of the structure determination and refinement are listed in Table 1 and Table 2. Low-temperature DSC studies showed a weak, reversible, and elongated endothermic heat effect (ESI Appendix A) with peak maxima at −36.5 °C (236.65 K) and −42.6 °C (230.55 K) and with 0.98 J/g and 1.58 J/g heat effects at 10 and 20 °C/min heating rates, respectively. The calculated powder XRD pattern of the SXRD measurement at 100 K agrees well with the experimentally found powder X-ray data measured at room temperature, and no distinguishable differences were found in the number of the calculated peaks from the single crystal X-ray data found both at 100 and 273 K (ESI Appendix A). The calculated peak positions completely coincided at 100 K and 273 K at low but were slightly shifted at high 2θ values, which may be attributed to the influence of thermal expansion. The phase transition, however, revealed that a process without changing the crystal symmetry proceeds, which resulted in an endothermic signal in the DSC curve (ESI Appendix A).

Symmetry codes to generate equivalent atoms:y + 2/3, −x + y + 1/3, −z + 1/3−x + y, −x + 1, z−x + 2/3, −y + 1/3, −z + 1/3 + 1−x + y, −x, z

Compound **1** crystallizes in the trigonal R-3m space group. The compound slowly decomposes under X-ray irradiation, even at 100 K, and fast at 273 K. Thus the crystallographic parameters of the compounds are given from the measurements made at liquid nitrogen temperature. The hexaamminecobalt(III) cation has regular octahedral coordination without any distortion (all six Co-N distances are 1.959(2) Å) (Figure 1a). In the asymmetric unit, there are 1/12th of the complex cation, 1/6th of a permanganate ion, 1/12th of a chloride anion, and 1/12th of a potassium central ion. The potassium ions are coordinated as central ions by six bridging permanganate anions and two chloride ions. The coordination around the potassium ion is rhombohedral (Figure 1b). Three oxygen atoms of a permanganate anion (O2) are coordinated to potassium ions and the fourth oxygen atom (O1) is not coordinated. Additionally, every permanganate ion is connected to three potassium cations. This bridging tridentate coordination mode of the permanganate ion, especially to a weak acceptor alkali metal ion, is unprecedented. Therefore, detailed spectroscopic studies of compound **1** were also performed (see below).

Every permanganate ion is coordinated with three potassium ions, making a polymeric network. Two chloride ions are coordinated to the potassium ions axially with a 3.433(1) Å bond length at 100 K and a 3.471 Å bond length at 237 K. The permanganate oxygen atoms are coordinated to the potassium equatorially with a six-fold symmetry. Only the O2 oxygen atoms are coordinated to the potassium; O1 oxygen atoms stay uncoordinated. The O2-K1-Cl1 angle is 64.99(4)° in the 100 K structure and 65.15(8) in the 273 K structure. The potassium coordination is very similar to that of a 18-crown-6 ether axially coordinating two chloride ions, but the coordinated oxygen atoms are more bent out of the equatorial plane compared to a crown ether complex [85,86]. The K⸱⸱⸱O and K⸱⸱⸱Cl bond distances were found to be very similar to the values found in compound **1** in chlorido-ligated 18-crown-6 potassium complexes such as (18-crown-6)(chlorido)potassium dihydrate [85] and catena-[bis(μ^2^-chlorido)(18-crown-6)potassium(I)-silver(I)] [86]. The two equivalent chlorido ligands are weakly-coordinated (d_K⸱⸱⸱Cl_ = 3.509 Å), and their arrangement is transoid. The hexaamminecobalt(III) cationic layers alternate with negatively charged layers that are thoroughly connected with coordinative covalent bonds (Figure 2). The bond lengths, angles, and torsions at 100 and 273 K in compound **1** are listed in Table 2.

The linear thermal expansion coefficients were calculated and found to be 1.765 × 10^−14^ m/K and 7.735 × 10^−14^ m/K in the direction of a/b and c axes, respectively. The elongation in the direction of the c-axis is >4-fold higher than in the direction of a/b axes, which suggests an increasing distance between the cationic and anionic layers (Figure 2) accompanied by a decrease in the strength of the N-H⸱⸱⸱Cl and N-H⸱⸱⸱O interactions. (The N⸱⸱⸱Cl and N⸱⸱⸱O distances between the ammonia ligands of the hexaamminecobalt(III) cations and the chloride ions, as well as the permanganate oxygen atoms, respectively, are given at 100 K and 273 K in Table 1). Based on these values, the weak endothermic effect on the DSC curve centered around −35 °C (ESI Appendix A) can be attributed to the changes in the secondary (hydrogen) bond interactions and not to a phase transition of compound **1**. The unit cell of compound **1** from the directions of the three different unit cell axes is shown in Figure 3.

The main parameters of the hydrogen bonds in compound **1** at 100 and 273 K are summarized in Table 3. Every oxygen atom of the permanganate ion and the chloride ions coordinated to potassium take part in hydrogen bond interactions. There are 6 × 5 = 30 hydrogen bond interactions and 12 N-H⸱⸱⸱Cl and 18 N-H⸱⸱⸱O interactions per monomeric unit. The weakest interaction was found between the chloride ion and one of the ammonia hydrogen atoms (C); the same ammonia hydrogen (C) also interacts with one of the coordinated bridging permanganate oxygen atoms. The second hydrogen of the ammonia (B) is coordinated both to a bridging and a non-bridging permanganate oxygen, whereas the third hydrogen (A) is coordinated to the chloride ion as well. Accordingly, two ammonia hydrogen atoms are bifurcated, whereas one hydrogen is monofurcated. Similarly, the bridging oxygen atoms and chloride ions have two hydrogen bond interactions each, whereas the non-bridging permanganate oxygen has only one hydrogen bond. The strongest hydrogen bond interaction can be found between the six ammonia ’B’ hydrogen atoms and six non-bridging permanganate oxygen atoms.

The hydrogen atomic positions are uncertain because the structures contain transition metals with high electron densities. Nonetheless, hydrogen atoms were generated in assumed geometries and included in structure factor calculations to better describe the real electron densities and to get a more realistic image in the Hirshfeld surface calculations. However, the hydrogen positions are not further discussed because of the uncertainty of their real coordinates. Thus, in our discussion of hydrogen bonds, only the data of the donor and acceptor atoms are considered. The hydrogen atomic positions are doubled by a mirror plane running through the ammonia nitrogen and the Co-N bond that resulting in six hydrogen positions with 50% occupancies each. Because of the uncertainty of the hydrogen atomic positions, we do not discuss the disorder of the ammonia hydrogen atoms.

Upon increasing the temperature from 100 K to 273 K, the elongation (weakening) of the hydrogen bonds is not equal in the five types of N-H⸱⸱⸱X (X = O, Cl) interactions. The smallest change (0.018/0.018 Å and 0.61/0.64%) belongs to the N-H⸱⸱⸱Cl interactions, whereas the longest ones belong to the coordinated oxygen atoms (0.025/0.025 Å and 0.80/0.80%), and the hydrogen bonds belonging to non-coordinated oxygen atoms change between these two values (0.022Å/0.74%). To find the changes in intermolecular interactions within the crystal structure at 100 K and 273 K, a Hirshfeld surface analysis was performed. The Hirshfeld surface method divides the volume of the crystal into two distinguishable regions: (1) where the sum of the electron densities contributed from a selected molecule(s) is larger, and (2) where the sum of the electron densities contributed from a selected molecule(s) is smaller than the contribution from other atoms(ions) located in the crystal. The distances of the nearest internal and external atoms to the surface, d_i_ and d_e_, respectively, and the sum of these two values (normalized by the corresponding van der Waals radii (d_norm_)), may be used to visualize the interactions within the crystal structures (Figure 4). The Hirshfeld surface analysis results (Figure 4 and Figure 5) of compound **1** at 100 K and 273 K showed that minor but defined changes occur in the hydrogen bond system and, accordingly, these small changes induce only a small endothermic heat effect in the DSC curve.

In general, the changes are minor; however, the relative areas of white (no interaction) and blue (weak interaction) are higher at 273 K than at 100 K, according to the weakening of all hydrogen bond interactions. The Hirshfeld surface analysis for the three kinds of interactions (H⸱⸱⸱H, H⸱⸱⸱O and H⸱⸱⸱Cl) can be summarized as follows: The area of color parts are 161.90 and 150.7 Å^2^ (165.9% and 158.9%), and the distributions of interactions between H-H, H-O and H-Cl interactions were found to be 27.40/46.00/26.60% and 25.60/45.90/28.50% at 100 K and 273 K, respectively.

### 2.2. Correlation Analysis of Compound ***1***

Experimental vibrational spectroscopic studies were performed at room temperature (IR, far-IR, and Raman), but Raman spectra were also recorded at liquid N_2_ temperature with 785 nm and 532 nm excitations (Figure 6 and ESI Appendix A).

The correlation analysis was performed by the “decomposition” of compound **1** into cationic (Co^3+^, K^+^), anionic (Cl^−^ and MnO_4_^−^), and neutral (NH_3_) structural elements (Figure 7 and Figure 8 and ESI Appendix A). There is only one type of permanganate anion in the crystal lattice of compound **1** (trigonal, Z = 3, the primitive cell contains 1 formula unit (Z/3)).

According to the C_3v_ site symmetry of the permanganate ion (the isolated permanganate ion has T_d_ symmetry), six internal Raman (A_g_(ν_1_,ν_3_,ν_4_) and E_g_(ν_2_,ν_3_,ν_4_) and IR bands (A_u_(ν_1_,ν_3_,ν_4_) and E_u_(ν_2_,ν_3_,ν_4_)) for each can be expected. The E_g_ and E_2u_ modes are doubly degenerate. Altogether, six bands due to internal MnO_4_^–^ vibrations are expected in the Raman (all of gerade symmetry) and six also in the IR (the latter are of ungerade symmetry). The symmetric stretching (ν_1_) and symmetric deformation (ν_2_) modes are active only in the Raman and IR spectra, respectively. Altogether, there are 18 vibrational degrees of freedom. All external gerade modes are Raman active, A_2u_ and E_u_ modes are IR active, whereas A_1u_ are IR inactive. A total of 12 vibrational degrees of freedom for the external modes (six for librations and 6 for hindered translations) can be expected. The correlation analysis results for the chloride ions are given in ESI Appendix A. The IR and Raman bands of the permanganate ion in compound **1** and their assignments are given in Table 4.

The coordination of the permanganate ions to potassium ions shifts the corresponding wave numbers as compared to the usual permanganate [87,88]. The doublet of the strong antisymmetric stretching Mn-O mode (ν_as_) consists of a high and a low-intensity component; the low-intensity component is located at 925 cm^−1^. The ν_1_ (ν_s_) symmetric stretching mode of the isolated tetrahedral permanganate ion is IR inactive and can become IR active due to distortion of the symmetry into C_3v_ [87,88]. However, this band is expected to be a weak one. The ν_s_(Mn-O) mode is the strongest band in the Raman spectrum of compound **1** and other permanganates [89] (Figure 7a,b); thus, the symmetric stretching Mn-O mode in the IR spectrum, which can be expected as a weak singlet in the IR spectrum, probably completely coincides with the wide rocking mode of the Co-NH_3_ fragments centered at 830 cm^−1^ (Figure 6c). The antisymmetric stretching and deformation modes appeared as doublets at −130 °C in the Raman spectra recorded with 532 nm excitation. The ν_as_(Mn-O) modes, however, are singlets at −80 °C and 25 °C. The resonance Raman effect was observed in the solid compound **1** at excitation of 785 nm, when, with decreasing intensity, the overtones of ν_1_(Mn-O) were observed at 1684 (2ν_1_), 2525 (3ν_1_) and 3363 (4ν_1_) cm^−1^ [89,90,91]. The external (translation) modes of two Cl^−^ anions (six vibrational degrees of freedom) consist of a normal singlet and a doubly degenerate singlet in both the IR and Raman spectra. The A_1g_ and E_g_ modes are Raman, whereas the A_1u_ and E_u_ modes are IR active. Compound **1** is very sensitive to laser excitation and decomposes fast, especially at irradiation with a 532 nm laser, even at −80 °C. No Raman bands were observed in the low-wavenumber region, which could be attributed to the K⸱⸱⸱Cl modes, but in the IR spectrum, at around 140 cm^−1^, the complex band may contain the vibrational mode belonging to K⸱⸱⸱Cl interactions, together with other translational modes [92].

### 2.3. Vibrational Spectroscopic Evaluation of Cation Modes in Compound ***1***

A correlation analysis of the cobalt(III) ion and ammonia ligands of hexaamminecobalt(III) cations in compound **1** revealed a series of ammonia ligand vibrational modes (six equivalent ammonia molecules in one symmetric octahedral CoN_6_ skeleton) (Figure 8). The correlation analysis results regarding the external (translation) mode of the potassium cation are given in ESI Appendix A. The band assignments of the isolated symmetric complex hexaamminecobalt(III) cation in compound **1**, based on normal coordinate analysis and quantum chemical calculations [38,42,93,94], are given in the Cif files of compound **1**.

There are 12 internal Raman and 12 internal IR modes, (4A_1g_, 2A_2g_, 6E_g_ and 2A_1u_, 4A_2u_, 6E_u_, respectively). The site symmetry is C_s_, and the E_g_ and E_u_ are doubly degenerate representations. There are 18 vibrational degrees of freedom for hindered translations and hindered rotations for the 6 NH_3_ molecules each (Figure 8). The hindered translations of one kind of Co^3+^ and K^+^ cations are IR active, while E_u_ is doubly degenerate. The antisymmetric and symmetric N-H stretching modes of the six equivalent ammonia ligands of the hexaamminecobalt(III) cation appeared as a singlet at 3267 and 3136 cm^−1^, respectively [38,42,93], in the IR spectrum. The antisymmetric and symmetric deformation modes were at 1600 cm^−1^ as a singlet and 1342/1320 cm^−1^ as a doublet, respectively. The rocking mode appeared as a medium-intensity band at 830 cm^−1^. The rocking mode (*ρ*(NH_3_)) positions of the coordinated ammonia ligands are sensitive to the strength of the hydrogen bonds in ammonia complexes [38,42,93], and compound **1** shows similar strength of hydrogen bonds mediated by the coordinated ammonia ligands, as in [Co(NH_3_)_6_]X_3_ complexes (X = Cl, Br). The symmetric deformation mode positions are also located near the d_s_ band of these hexaamminecobalt halide complexes. Accordingly, the relative Me-N bond strength parameters (ε), defined by Grinberg for the ammonia complexes [95], are also close to these parameters of [Co(NH_3_)_6_]X_3_ (X = Cl, Br). The N-H modes are very weak in the Raman spectra, but the CoN_6_ skeleton bands can be assigned in the Raman spectra recorded with 785 nm excitation (Figure 6b). The excitation at 532 nm caused decomposition, even at low temperatures (Figure 6a).

The regular CoN_6_ octahedron (O_h_) has six normal modes. Three of them, [ν_1_(CoN), A_g_], ν_2_(ν_as_(CoN), E_g_) and ν_5_(δ(CoN), F_2g_) are Raman, while two others (ν_3_(ν_s_(CoN), F_1u_) and ν_4_(δ_as_(NCoN), F_1u_)) are IR active modes. The ν_6_(δ(NCoN), F_2u_) mode is silent (inactive) in the IR and Raman spectra as well under its original octahedral symmetry. The band positions of this regular octahedral cation were determined based on normal coordinate analysis [38,42,93] (Table 5).

The regular nature of the octahedral skeleton, i.e., the two bands that appeared in the IR spectrum, could unambiguously be assigned as ν_3_ and ν_4_ modes [93]. The ν_1_ and ν_2_ modes are inactive in the IR, and singlets in the Raman spectra were recorded between −130 °C and room temperature with 785 nm excitation. The ν_5_ band did not appear in the Raman spectra of compound **1**. Altogether, there are 120 vibrational degrees of freedom in the vibrational spectra of compound **1**; 54 of these are due to internal vibrations of the MnO_4_^–^ anions (18) and the NH_3_ molecules (36), 24 are due to librations of the MnO_4_^–^ anions (6) and the NH_3_ molecules (18), and the remaining 42 are due to hindered translations. Three of the latter are due to acoustic modes, one with A_2u_ and one with E_u_ symmetry.

### 2.4. UV-VIS Study of Compound ***1***

The UV-VIS spectrum of the solid compound **1** is given in ESI Appendix A. The expected absorption bands [96,97,98,99,100,101] of four possible d-d transitions belong to [Co(NH_3_)_6_]^3+^ cations (Table 6), and the CT bands of the permanganate anions are partially overlapped [38,42,87]. The ground state of the low-spin central Co^III^ ion in [Co(NH_3_)_6_]^3+^ is t_2g_^6^ (^1^A_1g_). The excited electron (t_2g_^5^e_g_) spans with the ^3^T_1g_ + ^1^T_1g_ + ^1^T_2g_ + ^3^T_2g_ terms. The energies of the triplet states are lower than those of the singlets. The spin-allowed transitions (singlets) are expected to be weak bands [96,97,98,99,100,101]. The experimental UV-VIS bands are given in Table 6.

Two spin-allowed transitions (^1^*A*_1_→^1^*T*_1_ and ^1^*A*_1_→^1^*T*_2_) of the octahedral Co(NH_3_)_6_^3+^ cation were detected in the UV spectrum of compound **1**. The visible region of the UV-VIS spectrum of compound **1** consists of bands belonging to the CT bands of permanganate ions. The bands observed at 266 nm and 232 nm belong to the ^1^*A*_1_→^1^*T*_1_ and ^1^*A*_1_→^1^*T*_2_ cation and ^1^*A*_1_-^1^*T*_2_ (3*t*_2_-2*e*) and ^1^*A*_1_-^1^*T*_2_ (*t*_1_-4*t*_2_) permanganate ion transitions (259 nm and 227 nm for KMnO_4_, respectively) [2]. The ^1^*A*_1_-^1^*T*_2_ (*t*_1_-2*e*) transition of permanganate was assigned at 524 nm and 506 nm, whereas the bands at 485 nm and 568 nm (500 nm and 562 nm for KMnO_4_ in ref. [2]) may belong to both permanganate ions and cation transitions (Table 6). Similarly, the ^1^*A*_1_-^1^*T*_1_ (*t*_1_-2*e*) transition of the permanganate ion and the ^1^*A*_1_→^5^*T*_2_ transition of the hexaamminecobalt(III) cations also coincides (the permanganate bands were found at 720 and 710 nm for KMnO_4_ and [Agpy_2_]MnO_4_, respectively [7]).

### 2.5. Thermal Behavior of Compound ***1***

Compound **1** decomposes on heating with a strong explosion. The DSC curves (recorded in argon and O_2_ atmospheres) show that the first decomposition step (120 °C) is exothermic (ESI Appendix A), which reveals the direct role of atmospheric oxygen in initiating the decomposition processes. The simple ammonia ligand loss should be endothermic; thus, the exothermic reaction suggests the appearance of a redox reaction, as found in some other transition metal permanganate complexes containing ammonia ligands [3,15,16,17,18,102]. The second decomposition step consists of two and one part under inert and O_2_ atmospheres, respectively. Surprisingly, the reaction heat is greater in the Ar atmosphere than in O_2_. Due to the oxygen deficiency under the inert atmosphere (max. 8 oxygen atoms versus 6 NH_3_ ligands in compound **1**), and since the ammonia was formally the only reducing component (chloride ions may also be oxidized into chlorine by Co^III^ and permanganate ions, but the chlorine formed is reduced back into chloride ions by ammonia as a reducing agent; see below), the decrease of the reaction heat means that in addition to H_2_O as an oxidation product of ammonia with exothermic heat, some N-containing oxidation products having endothermic heat upon formation (NO or N_2_O, ΔH_f_^298^ = 90 and 82 kJ/mol, respectively) should form. The readily accessible external oxygen might facilitate the formation of these products, and the exothermic decomposition reaction heat found under the Ar atmosphere is suppressed due to the conversion of a larger part of NH_3_ into NO or N_2_O in O_2_ than under an inert atmosphere (ESI Appendix A). Since the decomposition temperatures are not changed due to this secondary oxidation process with gaseous oxygen, it can be assumed that the decomposition proceeds in the solid phase in both atmospheres, without the incorporation of the external oxygen, and that the gaseous intermediates formed are oxidized only by the external O_2_, which does not alter the decomposition temperature, but has a great influence on the reaction heat.

The TG curve of compound **1** in an inert atmosphere showed 43.9% overall mass loss, which corresponds to the loss of all six ammonia (as ammonia or its oxidized form) (ESI Appendix A) molecules and some oxygen- and chlorine-containing gaseous products, with the formation of KCl and CoMn_2_O_4_. The TG-MS studies confirmed the formation of HCl, H_2_O, and NO_x_ (ESI Appendix A). The solid decomposition intermediate that formed at 150 °C consists of an amorphous mixture and crystalline KCl. The amorphous Co and Mn containing residue with CoMn_2_O_4_ nominal composition started to crystallize at 250 °C (ESI Appendix A), which was completed at 500 °C (ESI Appendix A). The TG-MS studies performed under argon unambiguously showed the solid phase oxidation of ammonia into water by permanganate ions, because the hydrogen and oxygen in water may only be subtracted from ammonia and permanganate ions, respectively. The presence of HCl, as detected by TG-MS, might be attributed to a complex process consisting of:(a)the oxidation of chloride ions by Co^III^ into chlorine [38,41,42,103,104](b)the reaction of chlorine and ammonia to form HCl and N_2_(c)the reaction of HCl and ammonia into NH_4_Cl, and the dissociation of the sublimed ammonium chloride into NH_3_ and HCl in a vacuum, which were detected by TG-MS.

The IR spectrum of the decomposition residue prepared at 150 °C (below the temperature of the second decomposition step) confirmed the complete consumption of permanganate ions (ESI Appendix A). The IR spectrum of this decomposition intermediate contains bands that show the presence of N-H and N-O bond-containing species, together with some Co-O or Mn-O signals in the low wavenumber region. The aqueous leaching of this intermediate with subsequent evaporation to dryness resulted in a mixture. PXRD study of this dry material showed the presence of hydrated Co(NO_3_)_2_, KNO_3_, and KCl. No NH_4_NO_3_ peaks were detected (ESI Appendix A). The IR bands of the aqueous leachate showed the same but resolved bands as the unleached material, except for the low wavenumber metal–oxygen modes (ESI Appendix A). The observed bands belonged to the symmetric and antisymmetric stretching (3421, 3361, 3272, and 3197 cm^−1^) and deformation (1396 and 1609 cm^−1^, respectively) modes of the ammonium ion. The presence of ammonium ions was confirmed by the appearance of typical *ν*_2_ + *ν*_4_ combination bands at 2825 cm^−1^. The band at 1359 cm^−1^ belongs to the *ν*_as_(N-O) mode of the nitrate ion (ESI Appendix A). Since only the nitrate anion is present, but no NH_4_NO_3_ was detected, the only non-IR active anion to neutralize the charge of the ammonium ion, the ammonium salt of which is soluble in water, is the chloride ion. Thus, NH_4_Cl might appear in the mixture.

Regarding the anomaly in the DSC results, the TG-MS results of gaseous products were analyzed in detail. Water and ammonia evolved together in the inert atmosphere (*m/z* = 18, H_2_O^+^, and *m/z* = 15, NH^+^), due to the insufficient amount of oxygen in compound **1** to oxidize all NH_3_ into H_2_O and N_2_/NO_x_. Both NO and N_2_O were detected in the experiments done under argon or air atmosphere; however, their amount was found to be substantially higher in air than in an inert atmosphere, and the NO/N_2_O ratio was found to be >1 in argon, whereas that was <1 under oxidative atmosphere. These results confirm our assumption about the role of external oxygen in the formation of endothermic N-oxides in larger amounts in an air than inert atmosphere.

The presence of Co^II^, HCl, H_2_O, and N-oxides as oxidation products, together with the presence of nitrate ions and water, leads us to conclude that the first step of the decomposition is a solid phase quasi-intramolecular redox reaction. The initial decomposition intermediates decompose further in subsequent steps, resulting in the formation of KCl and CoMn_2_O_4_ (ESI Appendix A).

To prepare catalyst precursors, a safe decomposition route had to be found to decompose compound **1** without explosion. Thus, we studied its thermal decomposition under boiling toluene (b.p. 110 °C) to absorb the exothermic reaction heat by the evaporation of toluene and prevent the overheating and initiation of an explosion-like decomposition reaction. The decomposition product was studied by PXRD and IR (ESI Appendix A and Appendix A, respectively), then washed with water, and the aq. leachate was evaporated. Then, the crystalline phases were studied. The post-crystallization of the decomposition products was studied without aq. washing at 300 and 500 °C as well.

The decomposition reaction of compound **1** in toluene did not result in direct KCl formation (ESI Appendix A). However, the treatment of the decomposition residue with water resulted in the crystallization of KCl. This showed that potassium and chloride ions belonged to different species in the primarily formed intermediates, and KCl formed from them only after their dissolution in water. The other isolated salt from the aqueous leachate was cobalt(II) nitrate hexahydrate. The formation of NH_4_Cl and cobalt(II) nitrate (see above) in the reaction of Co^III^ and chloride ions in the presence of ammonia ensures an acidic NH_4_Cl solution during the aqueous leaching of the decomposition product, the reaction of which with a K-containing basic material resulted in the formation of KCl. The IR spectrum of the aqueous leachate showed that the decomposition in boiling toluene in 2 h was not complete, and thus, the reaction time had to be increased to 4 h.

The sample prepared under toluene was heated at 500 °C and a nanosized tetragonal CoMn_2_O_4_ spinel was found to be formed with 19 nm crystallite size, determined from the PXRD by the Scherer method. It was comparable with the crystallite sizes of the CoMn_2_O_4_ spinel prepared from compound **1** in the solid phase in the presence of air at 600 °C (25 nm) and with the size of the CoMn_2_O_4_ produced from the water-leached sample prepared under toluene with subsequent heating at 500 °C (23 nm). The metal–oxygen IR bands of the spinel structure appeared at 504 cm^−1^ and 491 cm^−1^ (ESI Appendix A). The SEM picture of the tetragonal CoMn_2_O_4_ spinel prepared at 500 °C with and without aqueous washing to remove KCl can be seen in Figure 9.

The long needle-like KCl crystals covering the spinel particles were removed completely by aqueous washing at room temperature, and the residue was the tetragonal CoMn_2_O_4_ with Co to Mn stoichiometry of 1:2. The catalytic activity of the phase pure and K-ion containing CoMn_2_O_4_ samples in Fischer-Tropsch and other fuel-producing reactions will be published in our ongoing paper dealing with the catalytic activity of various Co-Mn oxides with Co to Mn stoichiometry of 1:1-3 and prepared from various permanganate salts of ammonia-complexed cobalt as precursors.

## 3. Materials and Methods

The compounds containing reducing ligands and oxidizing anions (compounds **1**, **2** and **3**) are potential explosives; therefore, their handling requires great care.

Cobalt(II) chloride hexahydrate, 25% aq. ammonia, hydrogen peroxide, cc. HCl, ammonium chloride, activated carbon, deuterium oxide, and potassium permanganate were supplied by Deuton-X Ltd., Érd, Hungary.

Hexaamminecobalt(III) chloride was synthesized according to known methods [84,105].

Compound **1** was prepared as follows. 1.78 g (6.65 mmol) of [Co(NH_3_)_6_]Cl_3_ was dissolved in 10 mL of distilled water and heated to 60 °C. Then, a solution of 1.58 g (9.97 mmol) KMnO_4_ in 10 mL distilled water and heated to 60 °C was slowly added and stirred for five minutes. The solution was left to stand overnight at room temperature. The beautiful purple crystals were filtered off and washed with 50 mL cold (0 °C) distilled water and dried over freshly calcined (900 °C for 2 h) CaO in a desiccator for a day. The yield was 3.10 g (91.6%).

The conductivity of the saturated solution of compound **1** and the potassium permanganate, potassium chloride, and hexaamminecobalt(III) chloride solutions (at equivalent concentrations with the saturated compound **1**) were measured with a Jenway 4510 Conductivity meter (Cole Parmer Co., St. Neots, UK) equipped with a glass conductivity probe. The pH was measured with an AD111 pH meter (Adwa Instruments, Budapest, Hungary). The solutions were measured at 24.0 °C.

### 3.1. Single Crystal X-ray Diffraction

Violet platelet single crystals of K[Co(NH_3_)_6_]Cl_2_(MnO_4_)_2_ were measured at 100 K and 237 K, respectively. Intensity data were collected on a RAXIS-RAPID diffractometer (graphite monochromator, image plate detector) [106]. The structures were solved by direct methods [106] (and subsequent difference syntheses). Hydrogen atomic positions were calculated from assumed geometries. Hydrogen atoms were included in the structure factor calculation, but they were not refined. The isotropic displacement parameters of the hydrogen atoms were approximated from the *U*(eq) value of the atom they were bonded to.

The CCDC deposition numbers of compound **1** are 2362896 (100 K) and 2362897 (273 K).

### 3.2. Hirshfeld Surface Analysis

A Hirshfeld surface analysis was performed using CrystalExplorer 21.3. Hydrogen bond analysis was performed using Platon software (version 2023.1) [107].

### 3.3. Powder X-ray Diffraction

The powder XRD patterns of compound **1** were acquired at room temperature with the use of a Bragg-Brentano parafocusing goniometer (Philips Amsterdam, The Netherland), Cu Kα radiation, 1.5406/1.5444 Å, 2*θ* range of 4–70°, step size was 0.02°, 1 s interval time).

### 3.4. Vibrational Spectroscopy

Both the IR (Bruker Alpha, Bruker, Ettingen, Germany) and far-IR (Biorad, Budapest, Hungary) spectra of compound **1** were recorded in attenuated total reflectance (ATR) mode with 32 scans and a resolution of 2 cm^−1^.

The Raman spectra of the prepared materials were recorded from 2000 to 200 cm^−1^ using a Horiba Jobin-Yvon LabRAM microspectrometer (Horiba Co., Kyoto, Japan) with 785 nm diode laser source (~80 mW) and in the range of 4000–100 cm^−1^ with an external 532 nm frequency-doubled Nd-YAG (~40 mW). The equipment was coupled to an Olympus BX-40 optical microscope for focusing laser beams. Measurements were taken at both 298 K and 123 K, while the laser beam was focused through a 20× objective, and a D0.6 intensity filter was used, illuminating with 785 nm light to reduce the laser power to 25% to prevent degradation of the compounds. A D1 filter (~10% power) was applied during measurements with 523 nm excitation, as the compound was more sensitive to this energy. A confocal hole (1000 µm) and a monochromator with 950 and 1800 mm^−1^ groove gratings were utilized for light dispersion, achieving a resolution of 4 cm^−1^. Exposure times were set to 20–60 s to ensure peak intensity. Low-temperature Raman measurements at 123 K (−150 °C) were conducted using a Linkam THMS600 (Linkam Scientific Instruments, Salfords, UK), temperature control stage cooled by liquid nitrogen.

### 3.5. UV-Vis Spectroscopy

The UV-VIS diffuse reflectance spectrum (DRS) (Jasco V-670, NV-470 integrating sphere (Jasco, Tokyo, Japan) BaSO_4_ standard) of compound **1** was recorded between 200 and 800 nm at room temperature.

### 3.6. Scanning Electron Microscopy

The SEM pictures (JEOL JSM-5500LV(Nanyang University, Singapore) scanning electron microscope) of the decomposition product of compound **1** with and without aqueous washing were recorded with a sample specimen fixed on sample holders (Cu/Zn alloy, carbon tape), which were sputtered with a conductive Au/Pd layer for imaging. SEM studies were performed with a Zeiss EVO40 microscope (Carl Zeiss, Jena, Germany) operating at 20 kV.

### 3.7. DSC Studies

The DSC measurements above room temperature were done on a Perkin Elmer DSC 7 (Perkin Elmer, Shelton, CT, USA) instrument in an unsealed Al pan, with 3–5 mg sample mass and 5 °C/min heating rate under a continuous 20 cm^3^ min^−1^ argon or O_2_ flow. Low temperature DSC measurements of compound **1** were performed on a Setaram DSC92 (Lyon, France) system in sealed 120 μL aluminum pans, where the reference crucible was empty. Samples were scanned with a 10 °C/min heating rate from −140 °C to 30 °C. Raw measurements were blank corrected and data were processed with Calisto Processing v2.15 (AKTS, Valais, Switzerland).

### 3.8. Thermal Studies

The TG-MS measurements were carried out on a thermal analyzer (SDT Q600, TA Instruments, New Castle, DE, USA) coupled with a mass spectrometer (HPR20, Hiden Analytical, Cheshire, UK). The data points were collected from room temperature up to 600 °C in an inert (argon) and an oxidative (air) atmosphere, with flow rates = 50 mL/min. In the argon atmosphere, the heating rate was 10 K/min, but in air, it was reduced to 5 K/min and the sample was covered with powdered Al_2_O_3_ to prevent its explosion during the measurement. Sample holder/reference: alumina crucible/empty alumina crucible. Sample mass: 1.2–2.7 mg. Selected ions between *m*/*z* = 1–71 were monitored in multiple ion detection mode (MID).

The isotherm decompositions were performed in an air atmosphere in a tube with 5 °C/min heating rate and keeping the sample at the given temperature for 2 h. Similarly, 5 g of compound 1 was mixed with 100 mL of toluene, and heated slowly under reflux temperature. Then, the mixture was heated under stirring at reflux temperature for 4 h.

## 4. Conclusions

A double salt of KCl and the non-existing hexaamminecobalt(III) chloride dipermanganate, [Co^III^(NH_3_)_6_]_n_[(K(κ^1^-Cl)_2_(μ^2,2′,2″^-(κ^3^-O,O′,O″-MnO_4_)_2_)_n_^∞^] with unique structural features, including an unprecedented tridentate-bridging coordination mode of permanganate-ion and an eight-coordinated (rhombohedral) chlorido and permanganato ligated potassium ion with isolated regular octahedral hexaamminecobalt(III) cation, was prepared with a yield > 90%.

The crystal structure was determined; the anion is a potassium complex with a polymer structure stabilized by mono and bifurcated N-H⸱⸱⸱Cl and N-H⸱⸱⸱O (bridging and non-bridging) hydrogen bonds of undistorted hexaamminecobalt(III) cations.

Detailed spectroscopic (IR, far-IR, and Raman) studies resulted in the assignment of all normal modes, and the existence of the resonance Raman effect was also detected.

The thermal decomposition products of compound **1** at 500 °C were tetragonal CoMn_2_O_4_ spinel and KCl. The decomposition intermediates formed in toluene at 110 °C showed the presence of potassium and chloride containing intermediates, which combined into KCl during the aqueous leaching, together with the formation of cobalt(II) nitrate hexahydrate.

The solid phase redox reactions occurred at the first decomposition step, with the formation of Co^II^ and lower valence manganese oxides, together with nitrate ion containing intermediates.

## Figures and Tables

**Figure 1 molecules-29-04443-f001:**
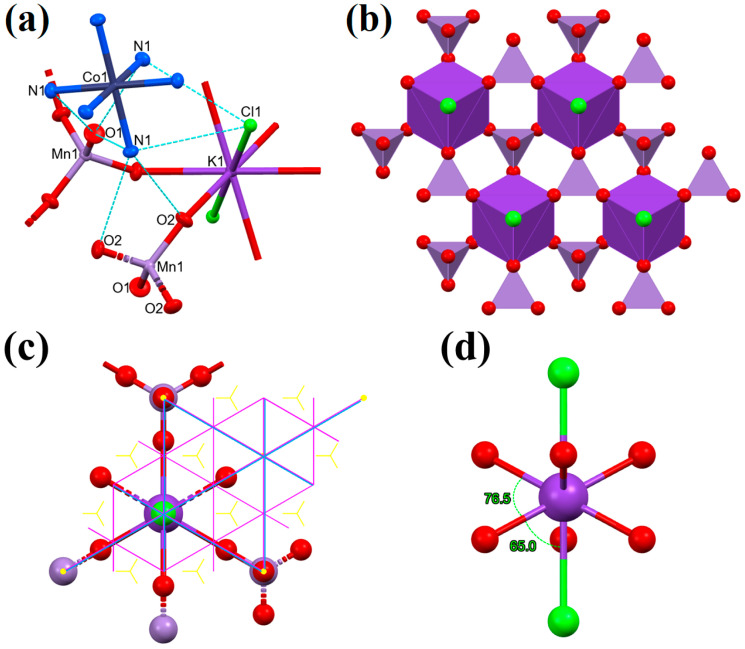
(**a**) Structure and labeling of compound **1**; hydrogen atoms are omitted for clarity (ellipsoid representation, thermal ellipsoids are drawn at the 50% probability level); hydrogen bonds are represented by light blue dashed lines. (**b**) Coordination of the anions around the potassium ions (polyhedral representation, hydrogen atoms are omitted for clarity; the potassium coordination sphere is represented by a dark violet polyhedron, permanganate anions by a light violet tetrahedron, and chloride ions by light green balls). (**c**) The coordination sphere of the potassium central ion (dark violet ball) showing the symmetry elements (glide planes: violet lines; mirror planes: light blue lines; three-fold screw axes: yellow triangles; three-fold roto-inversion axes: yellow dots), potassium is axially coordinated by chloride anions (green ball) and equatorially coordinated by permanganate ions (oxygen: red; manganese: light violet). (**d**) The potassium coordination sphere is shown from the direction perpendicular to the c axis with O-K-O and O-K-Cl bond angles.

**Figure 2 molecules-29-04443-f002:**
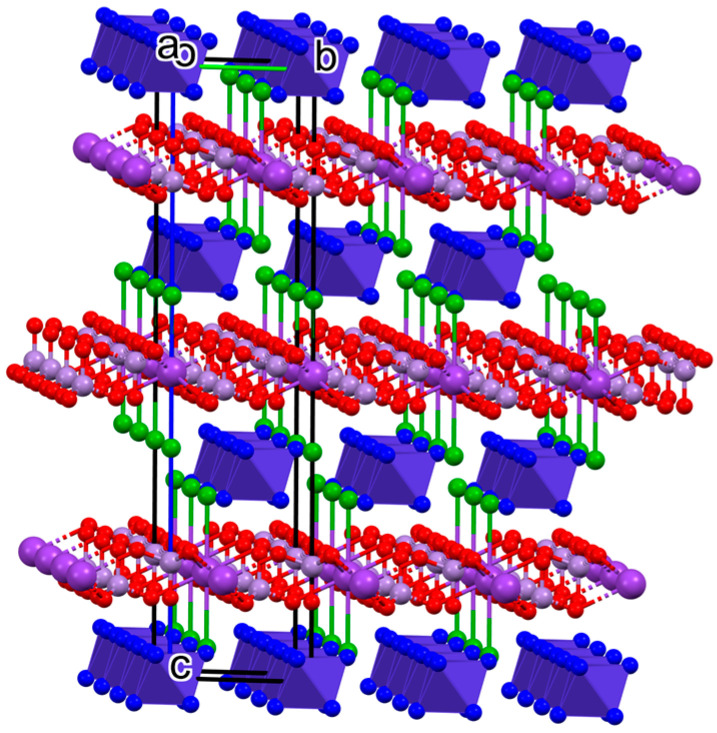
The unit cell of K[Co(NH_3_)_6_]Cl_2_(MnO_4_)_2_ (polyhedral and ball and stick representations; hydrogen atoms are omitted for clarity. Cobalt(III)hexamine complex cations are indicated by violet octahedra; potassium ions: dark violet balls; chloride ions: green balls; manganese: light violet balls; oxygen: red balls).

**Figure 3 molecules-29-04443-f003:**
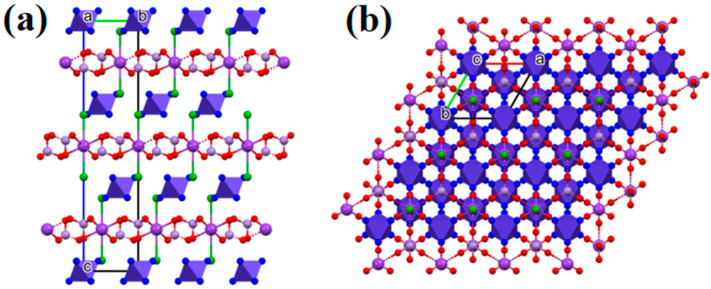
The packing arrangement in the crystal structure of compound **1** (**a**) in the direction of the a crystallographic axis and (**b**) in the direction of the c crystallographic axis (cobalt(III)hexamine complex cations are indicated by violet octahedra; potassium ions: dark violet balls; chloride ions: green balls; manganese: light violet balls; oxygen: red balls).

**Figure 4 molecules-29-04443-f004:**
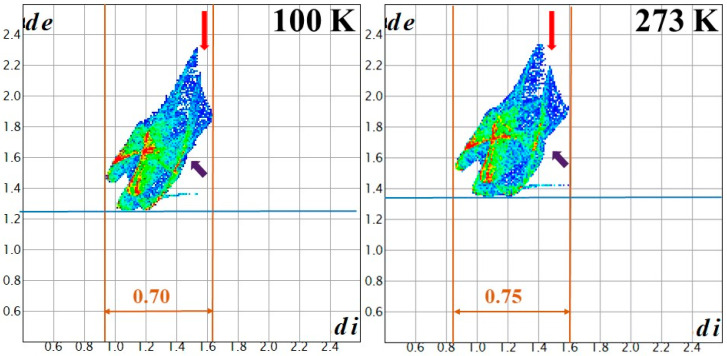
Hirshfeld surfaces of compound **1** at 100 K and 273 K. The change of the volume and the area of the hydrogen bonding interaction is the following: 161.9 and 150.7 Å^3^; 165.9 and 158.9 Å^2^ at 100 and 273 K, respectively. Arrows mean the place where changes were observed. The fingerprint plot colors are used to visualize these interactions, according to the relative area of the corresponding d_i_–d_e_ pair on the Hirshfeld surface: white—no contribution; blue—small contribution; green—medium contribution; red—large contribution. *d*_e_, the distance from the point to the nearest nucleus external to the surface, and *d*_i_ the distance to the nearest nucleus internal to the surface.

**Figure 5 molecules-29-04443-f005:**
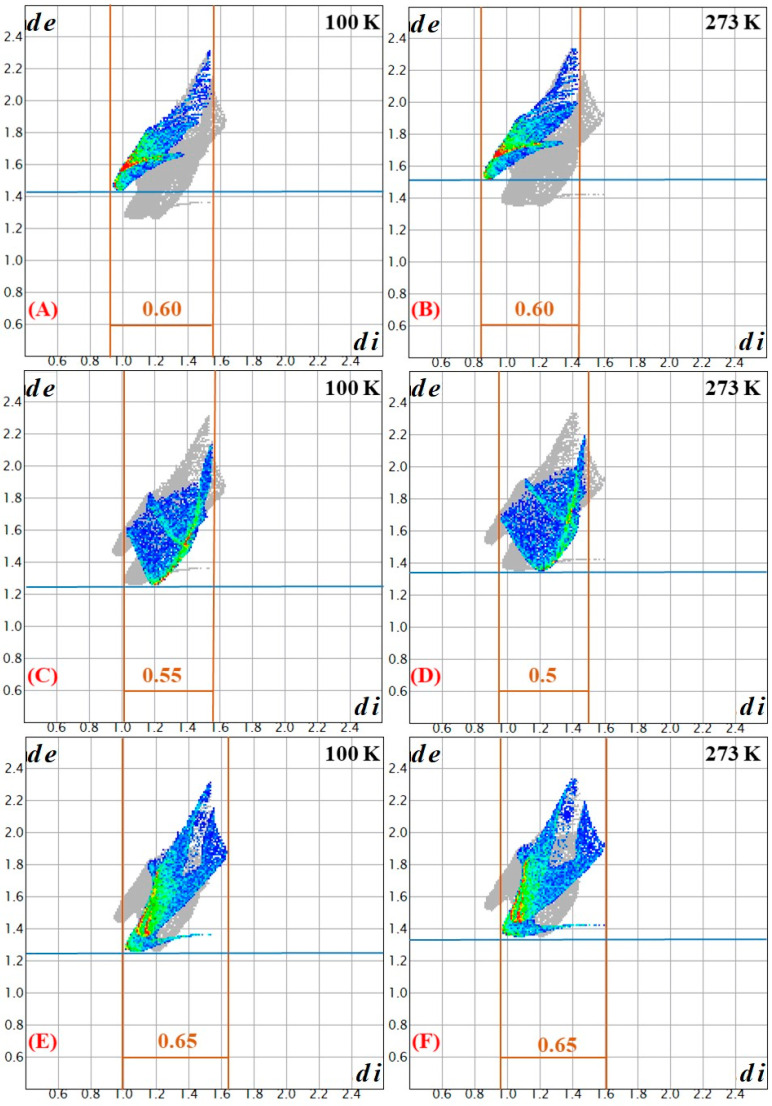
Hirshfeld surfaces of compound **1** at 100 K and 273 K (**A**,**B**) H-Cl; (**C**,**D**) H-H; (**E**,**F**) H-O. The change of the % of the specific hydrogen bonding interaction is the following: (**A**,**B**) H-Cl: 26.60 and 28.50% at 100 and 273 K, respectively; (**C**,**D**) H-H: 27.40 and 25.60% at 100 and 273 K, respectively; (**E**,**F**) H-O: 46.00 and 45.90% at 100 and 273 K, respectively. Arrows mean the place where changes were observed. The fingerprint plot colors are used to visualize these interactions, according to the relative area of the corresponding d_i_–d_e_ pair on the Hirshfeld surface: white—no contribution; blue—small contribution; green—medium contribution; red—large contribution. *d*_e_, the distance from the point to the nearest nucleus external to the surface, and *d*_i_ the distance to the nearest nucleus internal to the surface.

**Figure 6 molecules-29-04443-f006:**
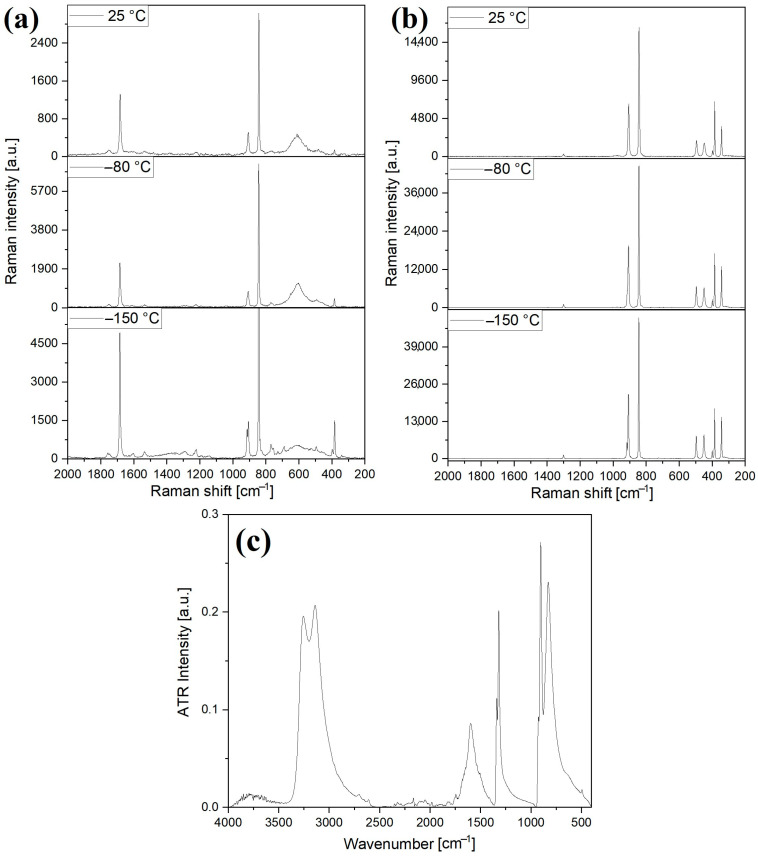
The Raman spectra (recorded with (**a**) 532 nm and (**b**) 785 nm laser) and (**c**) analytical range IR spectra (**c**) of compound **1**.

**Figure 7 molecules-29-04443-f007:**
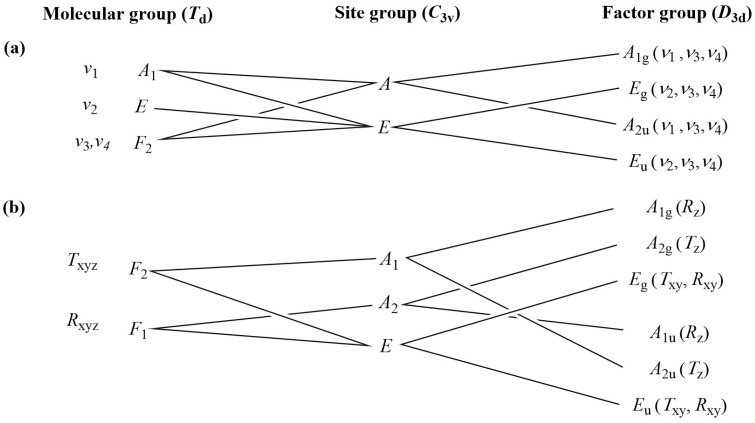
Correlation analysis results of permanganate ions (internal (**a**) and external (**b**) modes) in compound **1**.

**Figure 8 molecules-29-04443-f008:**
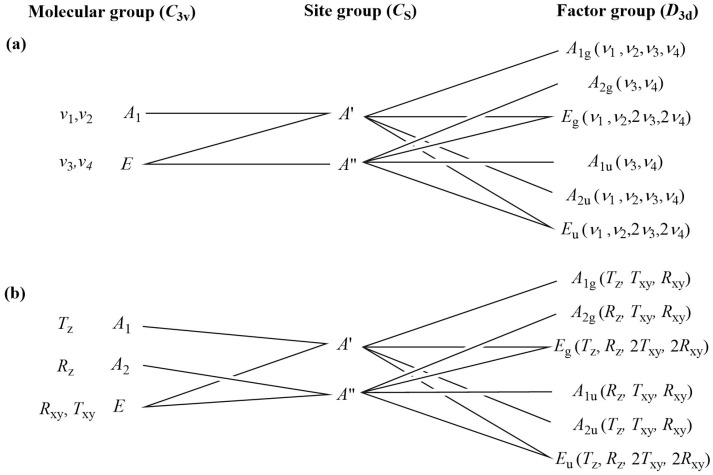
Correlation analysis of NH_3_ ligands (internal (**a**) and external (**b**) modes) in compound **1**.

**Figure 9 molecules-29-04443-f009:**
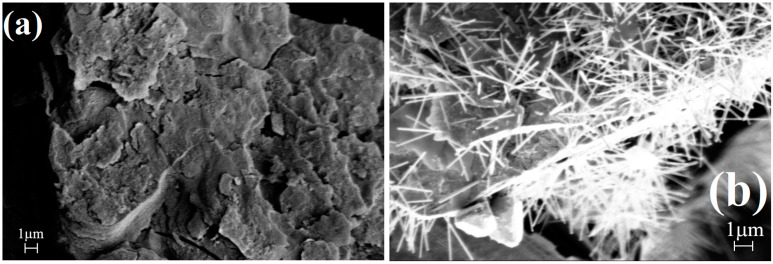
The SEM pictures of the final product of compound **1**, prepared at 500 °C (**a**) without and (**b**) with aqueous leaching.

**Table 1 molecules-29-04443-t001:** Crystal data and details of the structure refinement of compound **1**.

Empirical formula	Cl_6_Co_3_H_54_K_3_Mn_6_N_18_O_24_	
Formula weight	1527.04	
Temperature (K)	100 (2)	273 (2)
Radiation and wavelength (Å)	Mo-Kα, λ = 0.71073	Cu-Kα, λ = 1.54178
Crystal system	Trigonal	
Space group	R—3m	
Unit cell dimensions (Å)	*a* = 6.9642 (8)	*a* = 6.9969 (4)
	*c* = 27.6254 (18)	*c* = 27.7558 (12)
Volume (Å^3^)	1160.3 (3)	1176.78 (7)
*Z*	1	
Density (calculated, Mg/m^3^)	2.185	2.155
Absorption coefficient, *μ* (mm^−1^)	3.321	27.013
*F*(000)	762	762
Crystal colour	Violet	
Crystal description	Platelet	
Crystal size (mm)	0.64 × 0.60 × 0.07	0.33 × 0.28 × 0.17
Absorption correction	numerical	multi-scan
Max. and min. transmission	0.689/0.953	0.152/1.000
*Θ*-range for data collection	3.458 ≤ *Θ* ≤ 30.499°	4.780 ≤ *Θ* ≤ 68.131°
Index ranges	−9 ≤ *h* ≤ 9; −9 ≤ *k* ≤ 9; −39 ≤ *l* ≤ 39	−7 ≤ *h* ≤ 8; −8 ≤ *k* ≤ 8; −33 ≤ *l* ≤ 33
Reflections collected	18,492	13,017
Completeness to 2*Θ*	0.997	1.000
Friedel coverage	0.000	0.000
Independent reflections	489 [*R*(int) = 0.0619]	311 [*R*(int) = 0.1299]
Reflections *I* > 2*σ*(*I*)	489	305
Refinement method	full-matrix least-squares on *F*^2^	full-matrix least-squares on *F*^2^
Data/restraints/parameters	489/0/27	311/0/26
Goodness-of-fit on *F*^2^	1.452	1.249
Final *R* indices [*I* > 2*σ*(*I*)]	*R*1 = 0.0322, *wR*2 = 0.0798	*R*1 = 0.0585, *wR*2 = 0.1510
R indices (all data)	*R*1 = 0.0322, *wR*2 = 0.0798	*R*1 = 0.0593, *wR*2 = 0.1520
Max. and mean shift/esd	0.000; 0.000	0.000; 0.000
Largest diff. peak and hole (e·Å^−3^)	0.700; −0.935	1.057; −2.305

**Table 2 molecules-29-04443-t002:** Comparison of characteristic bond lengths (Å), angles (°), and torsions (°) at 100 K and 273 K.

Bond Lengths	100 K	273 K
Mn1—O1	1.604(4)	1.590(6)
Mn1—O2	1.614(2)	1.606(4)
Co1—N1	1.959(2)	1.950(4)
K1—O2	2.763(2)	2.791(4)
K1—Cl1	3.433(1)	3.471(2)
Bond Angles	100 K	273 K
O1—Mn1—O2	110.09(8)	110.2(1)
O2—Mn1—O2	108.85(8)	108.7(2)
Mn1—O2—K1	134.9(1)	135.0(2)
O2#1—K1—O2#2	76.59(5)	76.40(9)
O2#2—K1—Cl1#3	64.99(4)	65.15(8)
Torsion Angle	100 K	273 K
O2#4—Mn1—O2-K1	120.8(1)	120.9(2)

**Table 3 molecules-29-04443-t003:** Selected hydrogen bond interactions in compound **1**.

Nr	Donor--H⸱⸱⸱Acceptor	Symm. Op.	D⸱⸱⸱A, Å		
			100 K	273 K	Δ(A)/Δ(%)
1	N1—H1A⸱⸱⸱Cl1	*y*, *x*, −*z*	3.318(3)	3.336(5)	0.018/0.54
2	N1—H1B⸱⸱⸱O1	*x*, *y*, *z*	2.976(4)	2.997(6)	0.022/0.74
3	N1—H1B⸱⸱⸱O2	2/3 − *x,* 1/3 − *x + y*, 1/3 − *z*	3.138(3)	3.164(6)	0.025/0.80
4	N1—H1C⸱⸱⸱Cl1	*x*, −1 *+ y*, *z*	3.506(2)	3.524(4)	0.018/0.51
5	N1—H1C⸱⸱⸱O2	2/3 + *x* − *y*, 1/3 − *y*, 1/3 − *z*	3.138(4)	3.164(7)	0.025/0.80

**Table 4 molecules-29-04443-t004:** The IR and liq. N_2_ temperature Raman spectral data of the permanganate ions in compound **1**.

Band/Assignation	Compound 1		
	IR Wavenumber, cm^−1^	Raman Shift (748 nm Excitation), cm^−1^, −130 °C	Raman Shift (532 nm Excitation), cm^−1^, −130 °C
*ν*_1_(MnO_4_), *ν*_s_(*A*)	830 *	845	842
*ν*_2_(MnO_4_), *δ*_s_(*E*)	359	343	---
*ν*_3_(MnO_4_), *ν*_as_(*F*_2_)	925, 904	919, 907	914, 906
*ν*_4_(MnO_4_), *δ*_as_(*F*_2_)	395	400, 385	398, 383

* Coinciding band with *ρ*(Co-NH_3_).

**Table 5 molecules-29-04443-t005:** The IR and Raman spectral data of the CoN_6_ skeleton in compound **1**.

Mode	IR Wavenumber, cm^−1^	Raman Shifts, 785 nm Excitation, −130 °C
*ν*_1_(CoN_6_)*ν*_s_	-	497 *
*ν*_2_(CoN_6_)*ν*_as_	-	450
*ν*_3_(CoN_6_)*ν*_s_	503	-
*ν*_4_(CoN_6_)*δ*_as_	318	-
*ν*_5_(CoN_6_)*δ*_s_	-	not detectable
*ν*_6_(CoN_6_)*δ*	-	-

* Coinciding band.

**Table 6 molecules-29-04443-t006:** Wavelengths of the electronic transitions (in nm) of the hexaamminecobalt(III) cation in compound **1**.

Assignment in O Symmetry	Compound 1	[Co(NH_3_)_6_]Cl_2_MnO_4_ [42]	[Co(NH_3_)_6_]Cl_3_ [101]
^1^*A*_1_→^3^*T*_1_	Out of measurement range	830	833
^1^*A*_1_→^5^*T*_2_	708sh	727	730
^1^*A*_1_→^3^*T*_2_	602sh	-	617
	568 *	575sh	581
^1^*A*_1_→^1^*T*_1_	485sh	490	486
	455sh, 432	450	444
^1^*A*_1_→^1^*T*_2_	391	375	367
	345, 326sh	343, 330sh	324

* Coinciding bands.

## Data Availability

Data are contained with manuscript and Appendix A.

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
