# Peer review of "An Unprecedented Tridentate-Bridging Coordination Mode of Permanganate Ions: The Synthesis of an Anionic Coordination Polymer—[CoIII(NH3)6]n[(K(κ1-Cl)22,2′,2″-(κ3-O,O′,O″-MnO4)2)n]—Containing Potassium Central Ion and Chlorido and Permanganato Ligands"

_molecules, 2024, doi:10.3390/molecules29184443_

Round 1

Reviewer 1 Report

Comments and Suggestions for Authors

See pdf file

Author Response

Reviewer 2.

Thank you for evaluation of our paper. The answers are given below.

 Unfortunately, the authors' message, which is the leitmotif of the article and was even placed them in the title of the manuscript, contradicts the basic principles of coordination chemistry.

We absolutely disagree with this declaration because our compound is a typical coordination compound. 

Definition of coordination compound from https://www.britannica.com/science/coordination-compound:  Coordination compound, any of a class of substances with chemical structures in which a central metal  by nonmetal  atoms or groups of atoms, called ligands , joined to it by chemical bonds  atom  is surrounded.

Our reviewer used an old definition of coordination compounds cited in the Encyclopedia Britannica. 

According to the upper definition of a coordination compound, in system K[Co(NH3)6)][MnO4]22Cl, only the cation [Co(NH3)6)]3+ is the only coordinating element. However, the authors consider the rest of system X to be the coordinating polymer. The authors do not take into account that not any contacts of a metal with the neighboring atoms or ions can be considered as a chemical bond, since the latter has a limit length due to the fact that it must be ensured by the presence of a certain amount of electron density. As a matter of fact, in a single crystal of the studied system — K[Co(NH3)6)]2 MnO42Cl (1), positively charged layers, composed of coordination cations [Co(NH3)6)]3+ forming the hydrogen bonds (2.506 A) with chloride anions, are interspersed with the negatively charged layers, built from the tetrahedral permanganate anions and potassium cations (see Figure 1). The K—Cl  distances in the obtained compound 1 are 2.791 and 3.471 A, respectively. Taken into account the difference in temperature of the X-ray experiment, the value of 2.791 determined for K—O distance in 1 correlates well with the corresponding K—O distances of 2.913÷3.02 A found  in the crystal structure of KMnO4. However, the value of 3.471 для K—Cl in 1 is much lower compared to 3.147 known for potassium chloride (see Figure 2).

Our reviewer showed that there are ionic bonds in our compound, but it does not disclose the coordination nature of these ionic bonds. Taking the coordination bond as a dative covalent bond is an old conception, the new conceptions show that the coordination bond is mainly electrostatic (ionic) and rarely covalent. 

I am attaching some papers containing modern definitions of coordination bonds and the reasons for their “ionic” nature (see the crystal and ligand field theory). Furthermore, I gave some examples and atomic distances of crown-ether complexes – including the KCl – 18-crown-6 complex. They have K⸱⸱⸱O coordination bonds, which are mostly longer than, in our case, the K⸱⸱⸱O-(Mn) bonds. I think the reviewer accepts that the K-18-crown-6 complexes are coordination compounds. I attach some cif files, which show the structure of some chlorido-K-coordinated 18-crown-6-KX complexes with similar K⸱⸱⸱Cl distances as in our case, and Bikanina’s paper about coordination interactions in several hundred chloride ion-containing K compounds.

Then, following the authors' logic, who believe that the negative part of compound 1 is a two-dimensional coordination polymer {[(K(κ1-Cl)2(μ2,2’,2’’-(κ3-O,O’,O’’-MnO4)2)]3-}n, and also taking into account a vicinity of the values of the interatomic contacts K—O (and K—Cl), potassium permanganate is also a coordination polymer, as well as potassium chloride.

Figure 1. The view of crystal structure  of K[Co(NH3)6)][MnO4]22Cl (100).

Figure 2. The pseudo “coordination polymer” in crystal structure of potassium chloride.

Bikanina’s paper also shows that the “ionic” KCl has some coordination character (I3 type). However, the compounds in which the cations and anions are built in an infinite 3D ionic lattice as KCl or KMnO4 (Bikanina, Palenaik) are generally taken as ionic compounds in spite of coordination polyhedra around the K ion. In KCl, the lattice point contains discrete ions, which are regularly repeated with the neighboring opposite charged ions in 3D. Therefore, KCl is taken as an ionic compound due to this.  We have Cl⸱⸱⸱K⸱⸱⸱Cl linkage in our compound, which does not repeat in the axial direction, no other potassium neighbors of Cl ions. So it is not an infinite ionic lattice, but a lattice when instead of discrete ions, coordinative bonds containing elements are linked to each other, but only in 2 dimensions.

In addition, due to inattentiveness or poor knowledge of physical coordination chemistry, the authors treat Mn1 - K1 contacts as chemical bonds (see Tables S5 and S11).

The “bond distances” are given in Table form by the XRD software, and really, we were careless. It's our fault not to remove that, but let me reject the assumption that we wouldn't be aware of the basics of coordination chemistry.

The Supplementary Material is carelessly edited. Placing only one item (table or figure) on one page is a very unfortunate decision. Moreover, the pdf file was made starting from a file, in which not all revisions were accepted, so the resulting file has very wide margins when viewed.

The ESI has been revised. Thank you for calling our attention to this mistake. 

Considering that the greatest feature of the publication is that the unprecedented method of coordination of MnO4-, as well as the formation of a potassium coordination polymer is a fiction, and then scientifically the manuscript is of little value to be published in the general chemical journal "Molecules". Given that a crystalline phase of complicated composition has been  investigated in detail, a more targeted audience is the readers of the journ

Reviewer 2 Report

Comments and Suggestions for Authors

This  paper describes the synthesis and properites of an interesting polymer - [CoIII(NH3)6]n[(K(k 1 -Cl)2(m2,2’,2’’-(k 3 -O,O’,O’’-MnO4)2)n ∞] containing a central K cation and chloride and permanganate ligands.

Crystal structures of the complex at two different temperatures 103 and 273 are determined with structures with only minimal differences between them. The refinements are acceptable. However the coordinates of the atoms reported in the cif files are totally different in the two structures which is unacceptable. Those in the 273 structure are particularly strange, all atoms being place well outside the unit cell and these need to be changed to be similar to those in the 103 structure. In addition both cifs should contain details of the hydrogen bonds.

The paper contains far too many references. Nearly all of the excess are included in the introduction and only the most relevant should be included. 137 is far too many.

Table 1 is untidy. All units should be given in the first column and not in the second and third. Only independent cell dimensions are needed, thus just a and c lengths. For the 103 structure are there really 489 independent reflections all of which have I>2o(I)?  In this table it is stated that the 273 structure was measured with Mo radiation but this is not the case.  According to the cif file (and the enormous absorption mu). it is measured with Cu. Many of the details in this table are unnecessarily also included in the text (lines 487-519) and this section should be much shortened to avoid repetition. It is stated here that the H atoms were included in calculated positions but this is vague as  AFIX 137 is used in the 103 structure but AFIX 3 in the 273 structure. Therefore to compare the hydrogen bonds in the manner done in Table 2 is unjustified unless equivalent refinement techniques are used.  There is a further complication that is apparent in the cif files. N1 is bonded to 6 hydrogen atoms all with 50% occupancy with, three in x,y,z positions and three in  x, x-y, z positions so the treatment of the  H atoms and indeed the H bonds need more detailed discussion.

The description of the structure should be improved by mentioning the symmetry imposed on the atoms in the unitcell.  It is mentioned that the K coordination is rhombohedral but this is not an adequate description. the symmetry of the K site should be explained more clearly. Consider adding a figure showing the K coordination sphere. The dimensions of the bonds to K and Mn should be mentioned in the text with standard deviations.

As the a and b axes in this spacegroup are equivalent Figure 3b is the same as Figure 3a and should be omitted. In any case Figure 3a is essentially the same as Figure 2, though it is far clearer. The caption should contain the colour coding.

The description of H bonds in the text is hard to follow and is incomplete given that there are 6 positions for the H atoms around the N. A diagram showing the hydrogen bonds is necessary for clarification

The ESI needs to be improved. Table S1 is included in the main text and needs not be repeated here. Tables S2 to S4 are also unnecessary as their contents are in the cif files. Table S5 is necessary as it contains the molecular dimensions and could usefully be included in the main text. However as currently presented it is very poor. Only unique dimensions are required. This means only one Mn1-O2,  Mn1-K1, Co-N1 bond length. For angles these restrictions also apply only unique angles are required. Some of the angles included are clearly irrelevant, i.e. 28.76, many are given without standard deviations and 180.00 is obviously not needed. The authors should note that 89.1 and 90.9 are not independent of each other.

The Hirshfeld surface calculations are acceptable but are problematic when they are used to compare the 103 and 273 structures because of the different refinement techniques of the H atoms and the H disorder. Are all 6 H positions bonded to the N considered in the calculation and does this skew the results?

Further studies of the complex include spectroscopy, IR and Raman and correlation studies which are well carried out.

Providing the crystallography section can be improved this paper will be acceptable for publication.

Comments on the Quality of English Language

the english is good, only minor adjustments required

Author Response

We would like to thank the referee for the most relevant and valuable remarks concerning the representation of the refinement results.

This  paper describes the synthesis and properites of an interesting polymer - [CoIII(NH3)6]n[(K(k 1 -Cl)2(m2,2’,2’’-(k 3 -O,O’,O’’-MnO4)2)n ∞] containing a central K cation and chloride and permanganate ligands.

Crystal structures of the complex at two different temperatures 103 and 273 are determined with structures with only minimal differences between them. The refinements are acceptable. However the coordinates of the atoms reported in the cif files are totally different in the two structures which is unacceptable. Those in the 273 structure are particularly strange, all atoms being place well outside the unit cell and these need to be changed to be similar to those in the 103 structure. In addition both cifs should contain details of the hydrogen bonds.

The atomic coordinates in the 273 K structure have been harmonized with the coordinates in the 100 K structure, the atoms were moved inside the unit cell. BOND   $H command is used in both refinements. The new cif at 273 K has been uploaded to CCDC.  

The paper contains far too many references. Nearly all of the excess are included in the introduction and only the most relevant should be included. 137 is far too many.

We agree with our reviewer 3 about the number of references. Still, we had to revise papers by including new references many times to show the possible relations that belonged to the title compound in detail. However, we removed 28 references according to the reviewer’s instructions. 

Table 1 is untidy. All units should be given in the first column and not in the second and third. Only independent cell dimensions are needed, thus just a and c lengths. For the 103 structure are there really 489 independent reflections all of which have I>2o(I)?  In this table it is stated that the 273 structure was measured with Mo radiation but this is not the case.  According to the cif file (and the enormous absorption mu). it is measured with Cu. Many of the details in this table are unnecessarily also included in the text (lines 487-519) and this section should be much shortened to avoid repetition. It is stated here that the H atoms were included in calculated positions but this is vague as  AFIX 137 is used in the 103 structure but AFIX 3 in the 273 structure. Therefore to compare the hydrogen bonds in the manner done in Table 2 is unjustified unless equivalent refinement techniques are used.  There is a further complication that is apparent in the cif files. N1 is bonded to 6 hydrogen atoms all with 50% occupancy with, three in x,y,z positions and three in  x, x-y, z positions so the treatment of the  H atoms and indeed the H bonds need more detailed discussion.

The following discussion about the hydrogen atomic positions has been inserted into the text:

The hydrogen atoms are not clearly visible because the structures contain transition metals with high electron density. Nonetheless, hydrogen atoms were generated in assumed geometries and included in structure factor calculations to better describe the real electron densities and to get a more realistic image in Hirshfeld calculations. However, hydrogen positions were not further discussed because of the uncertainty of their real coordinates. Thus, only the data of the donor and acceptor atoms were considered in the discussion of hydrogen bonds. The hydrogen atomic positions of the ammonia ligands are doubled by a mirror plane running through the ammonia nitrogen and the Co-N bond that results in six hydrogen positions with 50 % occupancies each and further vagueness of the hydrogen positions.

Data in Table 1 have been rationalized.

There are 489 independent reflections in the 100 K dataset. The crystal quality was exceedingly good and the reflections intensive, all with I>2o(I). AFIX 137 was used also for the hydrogens in the 273 K structure for the better comparability.

The description of the structure should be improved by mentioning the symmetry imposed on the atoms in the unitcell.  It is mentioned that the K coordination is rhombohedral but this is not an adequate description. the symmetry of the K site should be explained more clearly. Consider adding a figure showing the K coordination sphere. The dimensions of the bonds to K and Mn should be mentioned in the text with standard deviations.

Two chloride ions are coordinated to the potassium ions axially with 3.433(1) Å bond length at 100 K and 3.471 Å bond length at 237 K. The permanganate oxygens are coordinated to the potassium equatorially with a six-fold symmetry. Only the O2 oxygens are coordinated to the potassium, O1 oxygens stay uncoordinated. The O2-K1-Cl1 angle is 64.99(4)° in the 100 K structure and 65.15(8) in the 273 K structure, respectively. The potassium coordination is very similar to that of an 18crown6 ether axially coordinating two chloride ions, but the coordinated oxygens are more bent out of the equatorial plane as compared to a crown ether complex.

The coordination of the potassium ions is shown in more detail in Figures 1c and 1d.

As the a and b axes in this spacegroup are equivalent Figure 3b is the same as Figure 3a and should be omitted. In any case Figure 3a is essentially the same as Figure 2, though it is far clearer. The caption should contain the colour coding.

Figure 3b has been omitted.

Figure caption: Packing arrangement in the crystal structure of compound 1 (a) in the direction of the a crystallographic axis and (b) in the direction of the c crystallographic axis (cobalt(III)hexamine complex cations are indicated by violet octahedra, potassium ions: dark violet balls, chloride ions: green balls, manganese: light violet balls, oxygen: red balls)

The description of H bonds in the text is hard to follow and is incomplete given that there are 6 positions for the H atoms around the N. A diagram showing the hydrogen bonds is necessary for clarification

There are only three symmetry-independent hydrogen atoms in the ammonia ligands.

The hydrogen atomic positions are unsure because of the presence of heavy elements like cobalt and manganese. Therefore, in Table 2, the hydrogen bond data are only given for the donor and acceptor atoms, and the hydrogen atomic positions are not discussed.

The ESI needs to be improved. Table S1 is included in the main text and needs not be repeated here. Tables S2 to S4 are also unnecessary as their contents are in the cif files. Table S5 is necessary as it contains the molecular dimensions and could usefully be included in the main text. However as currently presented it is very poor. Only unique dimensions are required. This means only one Mn1-O2,  Mn1-K1, Co-N1 bond length. For angles these restrictions also apply only unique angles are required. Some of the angles included are clearly irrelevant, i.e. 28.76, many are given without standard deviations and 180.00 is obviously not needed. The authors should note that 89.1 and 90.9 are not independent of each other.

The table listing the bond lengths, angles and torsion angles has been rationalized and included in the text. The unnecessary tables have been deleted from the ESI.

The Hirshfeld surface calculations are acceptable but are problematic when they are used to compare the 103 and 273 structures because of the different refinement techniques of the H atoms and the H disorder. Are all 6 H positions bonded to the N considered in the calculation and does this skew the results?

The hydrogen refinement methods in the two structures have been harmonized for better comparability of the Hirshfeld surfaces. Hirshfeld surface for the 273 K structure has been calculated again. No significant difference is observable between the original and the corrected surfaces.

All hydrogens were included in the Hirshfeld surface calculations. The symmetry-generated hydrogen atoms could not be excluded from the calculations. We hope these results are more reliable and closer to reality than what we could have gotten by deleting the hydrogen atoms.

Reviewer 3 Report

Comments and Suggestions for Authors

A very well performed thorough study on the synthesis and characterization of a K-Co-NH3-Cl-MnO4 compound! Congratulations. However, I have some very minor corrections/ comments, questions:

1)Table 1, empirical formula: it should read "Cl 6", not "C 16"

2) line 153: "the parameters" were determined at liquid N2 temperature. Which parameters?

3) Figure 2: I suggest that the explanation of the colour coding given in the caption to Figure 3 should also appear here. I find it very difficult to recognize where the K is situated.

4)Line 421: "ammonia was the only reducing compound", but in line 446 the Chloride is oxidized by Co(III) (and possibly by Mn(VII), too). Please clarify!

Author Response

Reviewer 3.

A very well performed thorough study on the synthesis and characterization of a K-Co-NH3-Cl-MnO4 compound! Congratulations. However, I have some very minor corrections/ comments, questions:

Thank you for your opinion and careful checking of our manuscript.  

  • Table 1, empirical formula: it should read "Cl 6", not "C 16"

It has been revised.

  • line 153: "the parameters" were determined at liquid N2 temperature. Which parameters?

The crystallographic parameters. The text has been revised

  • Figure 2: I suggest that the explanation of the color coding given in the caption to Figure 3 should also appear here. I find it very difficult to recognize where the K is situated.

It has been done.

  • Line 421: "ammonia was the only reducing compound", but in line 446 the Chloride is oxidized by Co(III) (and possibly by Mn(VII), too). Please clarify!

The reviewer is right, CoIII and permanganate ions oxidize chloride ions, but as we discussed, the oxidation product (chlorine) is immediately reduced back into chloride ion (as HCl) by ammonia. Thus, finally, the chloride-ion content does not change into another valence form of chlorine (KCl and NH4Cl were isolated), thus chloride formally does not act as a “real” reducing component. Thus the “only” reducing agent formally is ammonia, even if the chloride was oxidized (but re-formed). We improved the sentence, to show it. Thank you for calling our attention to it.  

Reviewer 4 Report

Comments and Suggestions for Authors

The authors revisited an old synthetic problem from 1887:

A double salt of KCl and the non-existent hexaammine cobalt(III) chloride dipermanganate has been synthesised with a yield of over 90%. Its unique structure is characterised by a tridentate-bridging coordination form of the permanganate ion and an eightfold coordinated potassium ion. The crystal structure consists of a polymeric potassium complex stabilised by hydrogen bonding of the undistorted hexaammine cobalt(III) cation. Spectroscopic studies confirmed the assignment of all normal modes and demonstrated the Raman resonance effect. Thermal decomposition at 500 °C led to the formation of CoMn2O4 spinel and KCl, while the intermediates formed at 110 °C exhibited potassium and chloride compounds and finally formed cobalt(II) nitrate hexahydrate. The work is well done and the experiments are carried out very carefully and the crystal structures are in detail described. Some minor points should be considered: e.g. page 8, lines 229-231:  "hydrogens", "oxygens", "chlorids" etc. should be replaced by "hydrogen atoms" etc

Comments on the Quality of English Language

Quality of English is suitable

Author Response

Reviewer 4.

The authors revisited an old synthetic problem from 1887:

A double salt of KCl and the non-existent hexaammine cobalt(III) chloride dipermanganate has been synthesised with a yield of over 90%. Its unique structure is characterised by a tridentate-bridging coordination form of the permanganate ion and an eightfold coordinated potassium ion. The crystal structure consists of a polymeric potassium complex stabilised by hydrogen bonding of the undistorted hexaammine cobalt(III) cation. Spectroscopic studies confirmed the assignment of all normal modes and demonstrated the Raman resonance effect. Thermal decomposition at 500 °C led to the formation of CoMn2O4 spinel and KCl, while the intermediates formed at 110 °C exhibited potassium and chloride compounds and finally formed cobalt(II) nitrate hexahydrate. The work is well done and the experiments are carried out very carefully and the crystal structures are in detail described. Some minor points should be considered: e.g. page 8, lines 229-231:  "hydrogens", "oxygens", "chlorids" etc. should be replaced by "hydrogen atoms" etc

Answer:

Thank you for your opinion and careful checking of our  manuscript.  

The listed mistakes have been revised.

Round 2

Reviewer 1 Report

Comments and Suggestions for Authors

See file pdf

Author Response

The only part of the obtained crystalline phase that can be attributed to a coordination species is a cation [Co(NH3)6]3+.

Answer

The reviewer repeats his previous position, that only the Co(NH3)6(3+) cation can be considered as a coordination compound in our complex. He does not give any logical reason why the polymer chain formed by the chloride and permanganate ligands around the potassium ion should not be a coordination compound. Obviously, it is not a simple ionic compound, but a coordination compound formed by ionic ligands, which is shown by the fact that the chloride ions are arranged axially below and above the potassium cation, at the same distance, in a way that the negative chloride ion is not surrounded by the positive ions as it should be in a non-coordinating but “purely” ionic compound.

I suppose that the authors of the article "Coordination compound" in this esteemed publication would be apparently disagreed with the opinion of the main author of the manuscript, as the content of the encyclopedia is regularly updated (see photos below).

Answer

The reviewer does not have the authority to take a stand on the question of what the authors of an encyclopedia article would think about our material. He/She doesn't have/can't have any information about what the authors of the encyclopedia would think about this specific issue.

I note that I previously explained in detail that this old definition does not conflict with the fact that our compound is a coordination compound. The encyclopedias contain short information without professional depth, they are intended for non-professionals (I also used Römpp’ chemical encyclopedia as a source, more than 40 years ago, as a child, but since then  I have been getting professional information from scientific articles containing more professional pieces of information).

Updating the encyclopedia entry does not mean that the latest currently used model is given, only that it has been recently updated - but still describes the basic definition from the 1920’s years (this is more than enough for non-professional users). I have to say again, any chemical bond (also ionic) can be between the central atom and the ligand, see Bethe's crystal field theory created in 1927, which discusses the coordination bond theory as an interaction between charged particles (ions).

 Without appropriate references, the authors' assertion is unsubstantiated

Answer

The reviewer himself stated that he does not thank the attached literature, because that is too much, and no any reviewer who has time to read so much. So, his/her statement that I didn't attach a reference, so my statement is “unsubstantial”, more than insulting, a net lie. Unacceptable from a reviewer.

I unfortunately cannot thank the correspondent author for providing the bibliography, as the file is so large (about 20 MB) that none of the reviewers has the time to thoroughly review. It would have been much better to simply provide a list of references with citations from them that have definitions of both coordination compound and coordination bonding.Here is small fragments from publications sent in huge cover-letter file.

None of these cited fragments does not contradict the notion of a classical coordination compound. It should be noted that the electronic structure of the complex, like the electronic structure of ordinary organic molecules, is described by molecular orbitals, each of which is a linear combination of atomic orbitals. To describe the structure of an ionic crystal or metal solids there are other approaches: topological analysis of the experimental and theoretical (fully periodic Hartree±Fock and density functional calculations) electron density (see for example a paper 10.1107/S0108767304015260) and band structure.

Answer 

The reviewer contradicts himself/herself, as he describes how many files I have attached, and he/she misses that I did not tell him where to find the definition. The definition right away on the first page of the attached book  - and I attached papers to confute the reviewer’s statements about KCl, KMnO4 and similar compounds he/she mentioned.   The definition he/she always  repeats  does not confute that our compound would be a coordination compound, it is only his/her statement.

I am not considering the { K 18-crown-6}+ fragment to be coordination compound. Such aggregates are study objects of supramolecular chemistry, and it is more logical to refer them to inclusion compounds.

Answer

 The compounds of crown ethers with alkali metal salts are typical coordination compounds, not inclusion compounds, as the reviewer stated.  I  attached hundreds of data on such compounds in a list, do their authors all have a bad idea of ​​what the coordination compounds are? All alkali metal crown ether compounds are listed in the Gmelin database with code 5 (coordination compound). According to them, everyone, except for the reviewer, misunderstands that these are coordination compounds ?

A slang has developed among crystallographers to call any positively charged ion existing in a structure as a coordination center when isolating polyhedra in crystal space, exemplified by the paper sent by authors (https://doi.org/10.1007/s11173-005-0013-6) and the following article - https://doi.org/10.1107/S0108767304015260

Answer

Does everyone use “slang” except the reviewer? Hundreds and thousands of authors?

The first time I had not noticed that Figure S2 is not mentioned in the main text of the manuscript

Answer
ESI Figure 2 has been given in line 107.

Reviewer 2 Report

Comments and Suggestions for Authors

The submitted Supplementary Material includes three cif files which do not belong to this work and should be omitted.

The authors have satisfactorily answered most of my queries and the paper is much improved but minor items need attention.

The temperature of data obtained for the low temperature work is still 100 in the text and 103 in the cif so which is correct?

Line 183 the distance should be written as 3.471(1)

The discussion of the refinement of the hydrogen atoms is confusing. The section in the text (lines 220-234) appears in the original version and is unchanged. However after my comments, the authors have added a new section which clarifies  the treatment of the hydrogen atoms. (lines 235-244).  These two section need to be combined with the second section concerning the refinement coming first

 The authors should make it clear that the HFIX 137 option in shelxl was used for the 100 structure to refine three hydrogen atoms as a rigid group and that inclusion of the three hydrogens significantly reduced the R value from 0.040 to 0.032 and this treatment converged successfully so their positions can be justified. Thus a disordered model with the three hydrogen atoms H1A, H1B, H1C given 50% occupancy as they are disordered over a mirror plane to give two different sets of three H atoms is proposed. Therefore it is not correct to say ”that the hydrogen positions are not further discussed because of the uncertainty of their real coordinates” Therefore it is desirable to include dimensions including H atoms in Table 3. If the hydrogen atoms in 273 could not be refined using HFIX then this should be stated and concluded that the hydrogen positions could not be confirmed.

Author Response

First of all, we would like to express our thanks for the comments, which helped to improve the quality of our manuscript.

The temperature of data obtained for the low temperature work is still 100 in the text and 103 in the cif so which is correct?

It has been done. 

Line 183 the distance should be written as 3.471(1)

It has been done.

The discussion of the refinement of the hydrogen atoms is confusing. The section in the text (lines 220-234) appears in the original version and is unchanged. However after my comments, the authors have added a new section which clarifies  the treatment of the hydrogen atoms. (lines 235-244).  These two section need to be combined with the second section concerning the refinement coming first.

It has been done. (marked with yellow)  

 The authors should make it clear that the HFIX 137 option in shelxl was used for the 100 structure to refine three hydrogen atoms as a rigid group and that inclusion of the three hydrogens significantly reduced the R value from 0.040 to 0.032 and this treatment converged successfully so their positions can be justified. Thus a disordered model with the three hydrogen atoms H1A, H1B, H1C given 50% occupancy as they are disordered over a mirror plane to give two different sets of three H atoms is proposed. Therefore it is not correct to say ”that the hydrogen positions are not further discussed because of the uncertainty of their real coordinates” Therefore it is desirable to include dimensions including H atoms in Table 3. If the hydrogen atoms in 273 could not be refined using HFIX then this should be stated and concluded that the hydrogen positions could not be confirmed.

It has been done. The table and the experimental part have been modified. (marked by yellow).